# *Arnica montana* L.: Doesn’t Origin Matter? [note 1]

**DOI:** 10.3390/plants12203532

**Published:** 2023-10-11

**Authors:** Thomas J. Schmidt

**Affiliations:** University of Münster, Institute of Pharmaceutical Biology and Phytochemistry (IPBP), PharmaCampus—Corrensstraße 48, D-48149 Münster, Germany; thomschm@uni-muenster.de

**Keywords:** *Arnica montana* L., herbal medicinal product, traditional use, sesquiterpene lactone, helenalin, 11α,13-dihydrohelenalin, chemotype, subsp. *montana*, subsp. *atlantica*, anti-inflammatory activity

## Abstract

*Arnica montana* L. (Asteraceae) has a long and successful tradition in Europe as herbal medicine. Arnica flowers (i.e., the flowerheads of *Arnica montana*) are monographed in the European Pharmacopoeia (Ph. Eur.), and a European Union herbal monograph exists, in which its use as traditional herbal medicine is recommended. According to this monograph, Arnica flowers (Arnicae flos Ph. Eur.) and preparations thereof may be used topically to treat blunt injuries and traumas, inflammations and rheumatic muscle and joint complaints. The main bioactive constituents are sesquiterpene lactones (STLs) of the helenanolide type. Among these, a variety of esters of helenalin and 11α,13-dihydrohelenalin with low-molecular-weight carboxylic acids, namely, acetic, isobutyric, methacrylic, methylbutyric as well as tiglic acid, represent the main constituents, in addition to small amounts of the unesterified parent STLs. A plethora of reports exist on the pharmacological activities of these STLs, and it appears unquestioned that they represent the main active principles responsible for the herbal drug’s efficacy. It has been known for a long time, however, that considerable differences in the STL pattern occur between *A. montana* flowers from plants growing in middle or Eastern Europe with some originating from the Iberic peninsula. In the former, Helenalin esters usually predominate, whereas the latter contains almost exclusively 11α,13-Dihydrohelenalin derivatives. Differences in pharmacological potency, on the other hand, have been reported for the two subtypes of Arnica-STLs in various instances. At the same time, it has been previously proposed that one should distinguish between two subspecies of *A. montana*, subsp. *montana* occurring mainly in Central and Eastern Europe and subsp. *atlantica* in the southwestern range of the species distribution, i.e., on the Iberian Peninsula. The question hence arises whether or not the geographic origin of *Arnica montana* flowers is of any relevance for the medicinal use of the herbal drug and the pharmaceutical quality, efficacy and safety of its products and whether the chemical/pharmacological differences should not be recognized in pharmacopoeia monographs. The present review attempts to answer these questions based on a summary of the current state of botanical, phytochemical and pharmacological evidence.

## 1. Introduction

*Arnica montana* L. (Asteraceae) is a medicinal plant species that has been used for many centuries in European medicine. From the Middle Ages onwards, Arnica was shown and mentioned in various old herbal books and gained importance as a remedy up to the 18th and early 19th century ([1] and original literature cited there). It is important to note that this review is exclusively confined to the use of Arnica as part of modern rational science-based medicine and that its—rather popular—use in alternative therapy systems such as homeopathy will *not* be taken into account.

*A. montana* grows in montane to alpine habitats of continental Europe, up to about 3000 m altitude, in nutrient-poor acidic soils. It is an herbaceous perennial, forming a short, unbranched rhizome, with aerial parts growing to 15–60 cm height. It has sessile, mostly ovate to lanceolate-shaped opposite leaves, most of which form a basal rosette, which typically bears one to three bright-yellow flowerheads, 6–8 cm in diameter, each consisting of numerous actinomorphic hermaphrodite disc florets surrounded by a single row of 11–15 (20) zygomorphic, female ray florets, on an upright, sparsely branched stem (see Figure 1) [2]. All parts of the plant have reportedly been used in phytomedicine ([1,3] and original literature cited there). While the leaves or total aerial parts (Arnicae folium, Arnicae herba) and the underground parts (Arnicae radix) are not frequently used nowadays, the flowerheads are presently in wide application and the only part of the plant monographed in the European Pharmacopoeia (Ph. Eur.) under the name Arnica flower—Arnicae flos [4]. Here, it is defined that the drug must originate from *Arnica montana* L. and not from any other of the approximately 30 existing *Arnica* species, some of which have reportedly been used in folk medicines around the Northern Hemisphere [5,6,7]. Please note that, henceforth, in this manuscript, the term “Arnica flowers” is used without italics and refers exclusively to flowers of the species *A. montana* L. in the sense of Ph. Eur. [4]. Consistently, the non-italic term “Arnica” refers to *Arnica montana* L. (and no other *Arnica* species) in the context of medicinal use or preparations, while it will be italicized where the systematic genus name is meant. In spite of its name, the drug Arnica flower consists not only of the flowers (disc and ray florets in this case) but comprises the complete inflorescence, including the receptacle as well as involucral bracts [3,4,8].

Although a protected species under the EU Habitats Directive and the EU regulation of trade of fauna and flora, *A. montana* is currently only listed in the IUCN red list as a species of “least concern” at the European level [9] so that, in spite of its status as an endangered species in some regions, flowerheads can still be collected at wild habitats in certain areas (see detailed regional information in [9]). The species is difficult to cultivate on an agricultural scale, but attempts to find a more easily cultivable clone with a good yield of flowerheads were successful in the 1990s, so that cultivar Arbo (for “Arnica Bomme”) [10] is available on the market.

Arnica flowers are widely used in preparations based on alcoholic extracts such as the ethanolic tincture, which is also monographed in the Ph. Eur. as Arnica tincture (Arnicae tinctura) [11]. Less frequently, extracts obtained with vegetable oils such as sunflower oil are also in use [1,8].

While Arnica was historically used internally as well as externally, the former was abandoned in the 20th century due to a certain degree of toxicity upon internal use [1,8,12,13,14,15,16,17]. Thus, modern use is restricted to external application on intact skin. Several scientific monographs dealing with the use of Arnica and/or its preparations have been issued in the last few decades, most prominently by the former German Commission E [14] and by the European Scientific Cooperative on Phytotherapy (ESCOP) [15]. In 2014, the Committee on Herbal Medicinal Products (HMPC) of the European Medicines Agency (EMA), based on its Final Assessment Report [16], issued a Community herbal monograph (CHM) [17] on “*Arnica montana* L., flos”, in which the legal status of Arnica and its preparations is defined: herbal preparations in semi-solid and liquid dosage forms for cutaneous use and produced on the basis of various defined ethanolic extracts are defined as herbal medicines in Traditional Use, meaning that they may be marketed after simplified registration. Traditional Use registration does not require clinical safety and efficacy trials but is accepted on grounds of sufficient safety data and plausible efficacy, mainly based on the literature, given that the drug has been in use for at least 30 years, including 15 years in the EU. In Germany, Latvia and Slovenia, certain Arnica preparations also have a market authorization due to “Well established use” [16]. The CHM also defines the medical conditions for which Arnica may be used: it is a traditional herbal medicinal product for the relief of bruises, sprains and localized muscular pain and in use in the specified indications exclusively based upon long-standing use [17].

From this backdrop, in the obvious presence of sufficient data, it might be expected that any Arnica preparation produced and used in accordance with the CHM should be equally efficacious and safe. “Case closed”, one might think. But some existing evidence on rather conspicuous differences in the chemical composition of the bioactive constituents of Arnica flowers originating from different parts of Europe happens to make things more complicated.

## 2. Botanical and Phytochemical Evidence

### 2.1. Botany of A. montana and Possible Segregation in Two Subspecies

The genus *Arnica*, as a whole, was comprehensively described by B. Maguire in his extensive monograph from 1943 [2]. *Arnica montana* L. was recognized there as the type species of the subgenus Montana (which comprises, in addition to *A. montana*, only one other species, the north-east American *A. acaulis*). *A. montana* is the sole *Arnica* species occurring in Europe south of Scandinavia. Its distribution was described, according to Maguire (citing the original description by Hegi), as “Europe, up north to northern France, Belgium, northwest Germany, Denmark, Scandinavia (Hustad, 63° N, south Norway), Pomerania, Western Prussia, Eastern Prussia, North and East Poland, Lithuania, Livonia, Courland; in south Europe (only in high altitude) up to Portugal, east and north Spain up to the Pyrenees, up to northern Italy, to the northern Balkan and south Russia” (translated from German by the present author). Maguire recognized the considerable polymorphy of *A. montana*, which had previously led to several attempts to segregate the species (e.g., to establish a separate species or variety from populations with somewhat petiolate leaves, i.e., *A. petiolata* Schur) but, obviously, he did not adopt the view. He explicitly mentioned that *A. montana* had not been segregated into any pronounced geographical populations [2]. Obviously, he was not aware, however, that at roughly the same time, Spanish botanist A. de Bolòs y Vayreda was about to postulate the existence of two *A. montana* subspecies, distinguished by morphological characteristics and delimited by their geographic origin, subspecies *montana* occurring in the central and east European ranges and subsp. *atlantica*, occurring only in the far (south) western range, i.e., on the Iberic peninsula with Portugal, Northern Spain and up to southwestern France [18]. The two subspecies were reported to differ morphologically, subsp. *atlantica* being less tall and more slender, with more lanceolate leaves and somewhat smaller flowerheads than subsp. *montana*. The existence of the two subspecies, thus proposed, has later been questioned, since the morphological characteristics/biometric data did not allow for a clear distinction [19]. The view of de Bolòs y Vayreda, thus, has not been commonly accepted. Recent work based on genetic comparison, however, has provided new evidence in favor of the existence of two subspecies. *A. montana* from the geographically distinct populations in Central/East Europe and such from Spain was genetically highly different. Schmiderer et al. [20] compared the microsatellite DNA of various populations from central western Europe (Germany, Austria, North Italy, even eastern French Pyrenees), on the one hand, with some from Spain, on the other, found to form well-separated clusters in the phylogenetic analysis, also correlating with phytochemical differences (see Section 2.2.3 below) so that de Bolòs y Vayreda’s recognition of the two subspecies was supported by their data. Similarly, Vera et al. [21] compared two polymorphic chloroplast DNA (cpDNA) markers in various *A. montana* populations from different habitats in Galicia (NW Spain). Their results also suggested the presence of two different genetic groups and were also congruent with two chemotypes described [21] (see Section 2.2.3). The cpDNA data obtained from a relatively limited number of Galician accessions were later refined by the same group, adding microsatellite data and including more accessions from a wider range of northern Iberian locations, confirming the existence of distinct population genetic units on the Iberian peninsula [22,23].

Very interestingly, phytochemical evidence has long been known to point in the same direction, i.e., the existence of two chemically and geographically distinct types of *A. montana*, as will be pointed out in Section 2.2.3.

### 2.2. Chemistry of A. montana and Existence of Two Chemotypes

#### 2.2.1. Chemical Constituents

The chemistry of the genus *Arnica* and of the title species in particular has been the subject of many studies. Overall, secondary metabolites of diverse chemical classes have been identified in the various *Arnica* species studied so far, and the sesquiterpene lactones (STLs) of several structural types are found as a predominant feature in most of them [24]. The constituents of *A. montana* have been reviewed several times [1,5,6,7,24]. The flower drug contains 0.2–0.8% STLs belonging to the helenanolide subgroup of pseudoguaianolides (i.e., 10α-methyl pseudoguaianolides), represented here by esters of helenalin (HEL) and 11α,13-dihydrohelenalin (DH), with various carboxylic acids, such as acetic, methacrylic, isobutyric, 2- and 3-methylbutyric as well as tiglic, angelic and senecioic acid [1,3,5,6,24,25,26]. In addition to these main constituents, the occurrence of a few guaianolides has also been reported [27]. The structures of the STLs known from *A. montana* are shown, and the abbreviations used further on for the various esters are explained in Figure 2.

In addition to STLs, the flowerheads contain essential oil consisting of sesquiterpenes, thymol derivatives and further monoterpenes, as well as a plethora of flavone and flavonol aglycones and their glycosides, polyacetylenes, caffeic acid derivatives, coumarins, carotenoids and fatty acids [1,3,8,12,24]. Furthermore, two (non-toxic) pyrrolizidine alkaloids [28], a diterpene [29] and various triterpene alcohols and esters [30] have been described as constituents.

#### 2.2.2. Quantitative Analysis of STLs

A variety of methods have been used and published for the analytical characterization of Arnica by means of its STLs. These methods have recently been reviewed [7].

It has been generally accepted for many years that the STLs are the constituents mainly responsible for the biological/pharmacological activity of *Arnica montana* flowers and their preparations (see below, Section 3), so it is important in the context of the present article to put some focus on quantitative data and its determination. The European Pharmacopoeia standardizes the herbal drug and the tincture to a minimum total content of STLs of 0.4% and 0.04%, respectively, calculated as 11α,13-dihydrohelenalin tiglate (DHTG) [4,11]. The method prescribed by Ph. Eur. for this purpose is an HPLC method using UV detection at 225 nm, exploiting the absorbance caused by the STLs’ α,β-unsaturated carbonyl chromophores and using another STL, α-Santonin, which is not a constituent of *A. montana*, as an internal standard. All peaks in the chromatograms corresponding to STLs are integrated, and the sum of their areas is converted to approximate STL content using the response factor of DHTG [4,11]. The response factor is a substance-specific correction factor, SCF, relating the detector response to the analytes with that of the internal standard (i.e., the signal intensity of Arnica STLs to that of α-Santonin). The Ph. Eur. method is based on an earlier one originally developed by Willuhn and Leven [26], who determined the correction factors for santonin and various STLs of the HEL and DH series. The correction factors determined by these authors (termed K_f_ in the original paper, see Table 1) were part of the denominator of the equation, i.e., the peak areas were *divided* by these K_f_ values, to calculate the STL content [26], while in the Ph. Eur., the value in its reciprocal form is now part of the numerator of the corresponding equation so that the peak area is *multiplied* by the SCF [4,11] (see Table 1); thus, the K_f_ value of DHTG initially reported [26] was 0.88, while a value of 1.187 is now used in the Ph. Eur. equation [4,11]. The reciprocal value of the former would actually be 1.136, but the present-day value was newly determined for the Ph. Eur. monograph, as was personally communicated to the author from the Ph. Eur. helpdesk (helpdesk@edqm.eu) on specific inquiry concerning this issue in August 2022. The term SCF, as referred to in the present publication, means a factor for use in the numerator as in the Ph. Eur. monograph, i.e., for multiplication with the STL peak area. Inspecting the reciprocal values of the K_f_ data determined and published by the original authors [26] in Table 1, it turns out that the HEL derivatives (probably due to their higher absorbance at the measuring wavelength caused by two enone chromophores) would all have 1/K_f_ = SCF values < 1, whereas in the case of DH derivatives, these values would be >1 in all cases where they could be determined. It thus follows that the simplistic method of calculating an “overall content of STLs, determined as DHTG” based on the SCF >1 of one particular DH ester would lead to realistic results only for other DH derivatives but would give results significantly too large for all derivatives of HEL, whose peak areas would actually have to be multiplied by a value < 1 in order to give a realistic value. The error of the Ph. Eur. method, i.e., the discrepancy of the results from the true content of STLs, will, thus, grow with the fraction of HEL derivatives in the overall mix. In other words, total STL contents determined with the Ph. Eur. method for Arnica samples containing more HEL than DH esters will yield results that are notoriously too high. An example to illustrate this problem is given in the footnote to Table 1. This obvious shortcoming will certainly have to be taken into account when quantitative data determined with the Ph. Eur. method for different chemotypes of *A. montana* (see below) are to be compared.

This source of systematic error should be kept in mind when comparing quantitative data of STL content in the Arnica drug and preparations determined with the Ph. Eur. method. Furthermore, it is important to note that the reported STL amounts from different studies can only be compared with some caution since different methods may lead to significantly different results. The various different methods of determination are, therefore, mentioned throughout the following section.

#### 2.2.3. Different STL Chemotypes of *A. montana* Occur in Distinct Regions of Europe

Most importantly in the context of the present communication, very conspicuous differences in the detailed profile of STLs in *A. montana* samples of Central/Eastern European and Spanish origins, apparently correlating with the subdivision proposed by de Bolòs y Vayreda [18], have been reported. In their comprehensive study on Arnica flowers in 1994, Willuhn et al. [25] reported on the STL content and qualitative composition of 39 different accessions of *A. montana* flowers originating from Central/East European (CEA; 35 samples) as well as Spanish (SPA; 4 samples) locations (see Figure 3 and Table 2). While the former generally contained a higher amount of HEL than DH esters, the latter showed an STL pattern consisting of DH derivatives to a much higher extent. The STL contents were determined via HPLC according to [26], the former calculated as HELIB, the latter as DHIV; thus, the shortcoming of the current Ph. Eur. method mentioned in Section 2.2.2 does not apply to the results of this work. The average ratio of HEL/DH derivatives (RHD) was 4.7 ± 2.7 for the CEA and 0.08 ± 0.09 for the SPA samples. The total contents of STLs (TCS) were 0.49 ± 0.18% in CEA and 0.68 ± 0.25% in SPA, respectively, so that there appeared to be no significant difference in this regard. It is noteworthy that the CEA accessions came from geographical locations covering a wide area from northwest Germany to the southern Alps and a wide range of altitudes, from sea level (island of Sylt, North Germany) via the low mountain range of Western Germany to the highest ones in the central and southern alps. Without exception, these accessions all showed an RHD > 1, mostly >>1, with a maximal value of 13.3 in the accession from Grödner Joch, Southern Tyrol, Italy. Interestingly, however, four out of seven CEA accessions from the Engadin area (Switzerland) displayed much lower RHD values close to 1. However, no clear correlations of STL content or RHD with geographical origin or altitude are obvious among the 35 CEA accessions of this study [25]. Similar patterns of STLs in CEA, always with a dominance of HEL derivatives, were later reported by various other authors. Thus, Clauser et al. [31], using a somewhat poorly described HPLC-MS method for quantification, reported on the STL content of 14 *A. montana* accessions from North Italian provinces, Trento, Brescia and Bergamo, growing at altitudes between 1272 and 2060 m, which all showed a predominance of HEL over DH derivatives, the highest RHD being 7.3, the lowest, however, being 1.2 and, thus, close to equal concentrations. No correlation with either geographical coordinates or with harvesting date or altitude of the habitat are observed in the presented data [29]. In a study on 10 German *A. montana* accessions ranging from the very north to the Bavarian Alps, Seemann et al. [32] found no correlations between STL pattern (determined via a capillary GC method also developed by Willuhn and Leven [26]) and various environmental parameters. All investigated accessions showed the HEL-dominated STL pattern typical of CEA, with the average RHD ranging from 2.5 to 10.6 and no obvious correlation of this ratio with any of the investigated parameters, including altitude [32]. A similar result, with no correlation between STL content and altitude, was obtained in a study on variation in STLs and phenolic metabolites in flowerheads of *A. montana* cultivar Arbo, known to present an HEL chemotype, grown under controlled conditions in various proving fields at different altitudes (590–2230 m) near Innsbruck, Austria [33]. An HPLC method using α-Santonin as the internal standard, obviously related to the Ph. Eur. method but not fully specified, was used by these authors for the quantification of STLs.

Importantly, a correlation of the two different chemotypes with genotypic differences between CEA and SPA (or subsp. *montana* and *atlantica*, respectively) was confirmed in the above-mentioned study on chloroplast DNA, in which the two different chemotypes were assigned based on HPLC analyses according to Ph. Eur. but without reporting detailed results of these analyses [20].

The locations of the various accessions of the two chemotypes as reported in the literature [25,31,32,34,35,36] are mapped in Figure 3. Their exact content of STLs of the HEL and DH types is reported in Table 2.

**Figure 3 plants-12-03532-f003:**
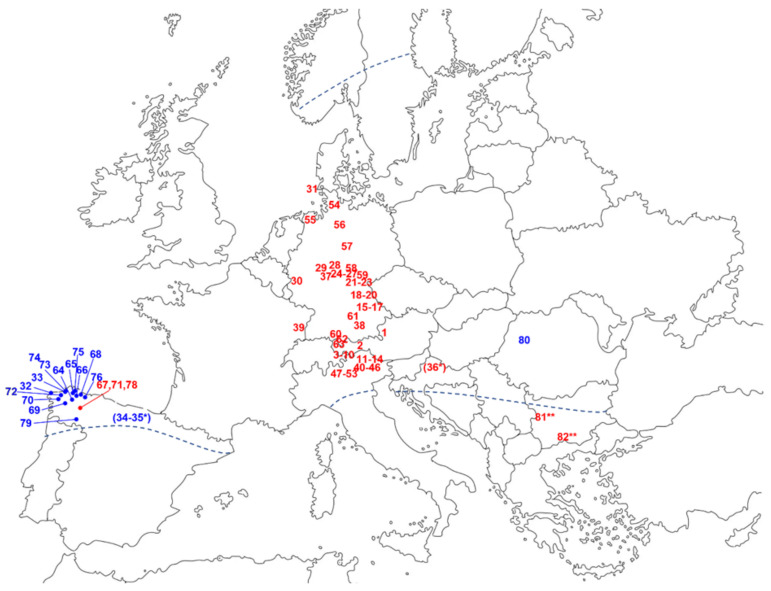
Distribution of *A. montana* and its chemotypes. The dashed lines indicate the approximate range of distribution of the species according to Maguire [2]. Note that, though not mentioned by this author, *A. montana* is absent from British and Irish Isles. Numerals denote the approximate locations of the various accessions mentioned in Table 2. Red color indicates an RHD > 1 (HEL chemotype) and blue color an RHD < 1 (DH chemotype). * Accessions 34–35 were from Spain and accession 36 from the former Yugoslavia but without specification of the exact location [25]. ** Accessions 81 and 82 were plants cultivated from seeds originating from Ukraine, Germany and Austria [36].

**Table 2 plants-12-03532-t002:** *Arnica montana* accessions with available data on the occurrence of HEL- and DH-type STLs and their reported locations of origin. The accession numbers are mapped in Figure 3. The RHD values are colored red if ≥1 (HEL chemotype) and blue if <1 (DH-chemotype), like in Figure 3.

No	Region	Country	Location	Total STL (%)	HEL-Der. (% of Total)	DH-Der. (% of Total)	RHD	Ref.
**1**	Alps	AT	Lungau	0.7	84	16	5.3	[25]
**2**	Alps	AT	Ötztal	0.2	88	12	7.3	[25]
**3**	Alps	AT	Silvretta, Fimbertal	0.3	84	16	5.3	[25]
**4**	Alps	CH	Engadin, Alpe Laretz	0.5	51	49	1.0	[25]
**5**	Alps	CH	Engadin, Tuoi-Hütte	0.6	52	48	1.1	[25]
**6**	Alps	CH	Engadin, Jantal-Hütte	0.7	59	41	1.4	[25]
**7**	Alps	CH	Engadin, Wiesbdadener Hütte	0.7	57	43	1.3	[25]
**8**	Alps	CH	Engadin, Ftan Pitschen	0.3	74	26	2.8	[25]
**9**	Alps	CH	Engadin, Scuol	0.5	72	28	2.6	[25]
**10**	Alps	CH	Engadin, Prada da Tuoi	0.4	78	22	3.5	[25]
**11**	Alps	IT	Regensburger Hütte	0.5	84	16	5.3	[25]
**12**	Alps	IT	Wolkenstein Campinoi	0.6	88	12	7.3	[25]
**13**	Alps	IT	Ref. Fermada	0.6	91	9	10.1	[25]
**14**	Alps	IT	Grödner Joch	0.5	93	7	13.3	[25]
**15**	Bayer. Wald	DE	Haidmühle	0.6	88	12	7.3	[25]
**16**	Bayer. Wald	DE	Hinterfirmiansreuth	0.4	83	17	4.9	[25]
**17**	Bayer. Wald	DE	St. Engelmar	0.2	87	13	6.7	[25]
**18**	Oberpf. Wald	DE	Gibacht b. Furth i.W.	0.5	81	19	4.3	[25]
**19**	Oberpf. Wald	DE	Schönau	0.3	85	15	5.7	[25]
**20**	Oberpf. Wald	DE	Silberhütte	0.3	69	31	2.2	[25]
**21**	Fichtelgeb.	DE	Kirchlamitz	0.6	83	17	4.9	[25]
**22**	Fichtelgeb.	DE	Weißenstadt	0.7	79	21	3.8	[25]
**23**	Fichtelgeb.	DE	Tröstan	0.4	76	24	3.2	[25]
**24**	Rhön	DE	Wüstensachsen	0.6	80	20	4.0	[25]
**25**	Rhön	DE	Wasserkuppe	0.4	77	23	3.3	[25]
**26**	Rhön	DE	Grabenhöfchen	0.5	72	28	2.6	[25]
**27**	Rhön	DE	Heidelstein	0.3	76	24	3.2	[25]
**28**	Meißner	DE	Meißnerhaus	0.4	82	18	4.6	[25]
**29**	Westerwald	DE	Rennerod	0.2	87	13	6.7	[25]
**30**	Eifel	DE	Baasern	0.4	82	18	4.6	[25]
**31**	North Sea Island	DE	Sylt	0.3	91	9	10.1	[25]
**32**	Galicia	ES	Berdoias/Muxia	0.6	12	88	0.1	[25]
**33**	Commercial *	ES	Herborista Mordage, La Coruna, Spanien	0.4	15	85	0.2	[25]
**34**	Commercial *	ES	Caelo GmbH, Hilden; (origin Spain) 1986	0.7	0	100	0.0	[25]
**35**	Commercial *	ES	Caelo GmbH, Hilden; (origin Spain) 1990	1.0	2	98	0.0	[25]
**36**	Commercial *	(YU)	Caelo GmbH, Hilden; (origin: former Yugoslavia)	0.6	76	24	3.2	[25]
**37**	Cultivated	DE	Exp. Station Rauischholzhausen	0.4	85	15	5.7	[25]
**38**	Cultivated	DE	Bayer. Landesanstalt f. Bodenkultur und Pflanzenbau, Freising	0.9	78	22	3.5	[25]
**39**	Cultivated	FR	Agricultural research institute Colmar	0.9	80	20	4.0	[25]
**40**	Alps	IT	Baito Casere Vece	1.2	76	24	3.2	[31]
**41**	Alps	IT	Malga Caserine di Dentro	1.3	65	35	1.9	[31]
**42**	Alps	IT	Malga Casina	0.9	80	21	3.9	[31]
**43**	Alps	IT	Malga Fregasoga	2.3	71	29	2.4	[31]
**44**	Alps	IT	Malga Juribello	1.0	84	16	5.2	[31]
**45**	Alps	IT	Malga Ora	1.1	55	45	1.2	[31]
**46**	Alps	IT	Malga Sass	0.5	71	29	2.5	[31]
**47**	Alps	IT	Malga Vericolo	0.8	67	34	2.0	[31]
**48**	Alps	IT	Monte Bondone	0.7	63	38	1.7	[31]
**49**	Alps	IT	Monte Peller	0.9	72	28	2.5	[31]
**50**	Alps	IT	Passo Campelli	0.8	67	33	2.0	[31]
**51**	Alps	IT	Passo Manghen	1.8	68	32	2.1	[31]
**52**	Alps	IT	Rifugio Bedole	1.2	66	34	1.9	[31]
**53**	Alps	IT	Rifugio Camini	1.5	88	12	7.3	[31]
**54**	Schleswig-Holstein	DE	Aukrug	0.9	n.a.	n.a.	8.1	[32]
**55**	Lower Saxony	DE	Tergast	0.6	n.a.	n.a.	4.7	[32]
**56**	Lower Saxony	DE	Unterlüß	0.8	n.a.	n.a.	5.8	[32]
**57**	Lower Saxony	DE	Braunlage	0.7	n.a.	n.a.	8.9	[32]
**58**	Thuringia	DE	Vesser	0.7	n.a.	n.a.	10.6	[32]
**59**	Bavaria	DE	Teuschnitz	0.6	n.a.	n.a.	8.5	[32]
**60**	Bavaria	DE	Immenstadt	1.1	n.a.	n.a.	2.8	[32]
**61**	Bavaria	DE	Schnellers	1.1	n.a.	n.a.	2.5	[32]
**62**	Bavaria	DE	Sonthofen	0.9	n.a.	n.a.	3.9	[32]
**63**	Bavaria	DE	Fellhorn	0.7	n.a.	n.a.	4.7	[32]
**64**	Galicia	ES	A Balsa	1.8	10	90	0.1	[35]
**65**	Galicia	ES	Aborbó	1.5	11	89	0.1	[35]
**66**	Galicia	ES	Aldixe	1.6	4	96	0.0	[35]
**67**	Galicia	ES	Alto do Couto	1.4	72	28	2.6	[35]
**68**	Galicia	ES	Campo do Oso	1.8	10	90	0.1	[35]
**69**	Galicia	ES	Chaos	1.4	21	79	0.3	[35]
**70**	Galicia	ES	Cruz da Golada	1.2	6	94	0.1	[35]
**71**	Galicia	ES	Formigueiros	1.2	57	43	1.3	[35]
**72**	Galicia	ES	Meira	1.7	9	91	0.1	[35]
**73**	Galicia	ES	Penchaina	1.9	5	95	0.1	[35]
**74**	Galicia	ES	Ponte Pedrido	1.7	4	96	0.0	[35]
**75**	Galicia	ES	Ponte Xestido	1.5	12	88	0.1	[35]
**76**	Galicia	ES	Reibocha	1.7	4	96	0.0	[35]
**77**	Galicia	ES	Sto Tomé	1.5	25	75	0.3	[35]
**78**	Galicia	ES	Sufoio	1.1	48	52	0.9 ^a^	[35]
**79**	Galicia	ES	Valdin	1.3	7	93	0.1	[35]
**80a**	Carpathians	RO	Apuseni mountains/silic	0.8	26	74	0.4	[34]
**80b**	Carpathians	RO	Apuseni mountains/calc	1.2	29	71	0.4	[34]
**81a**	Cultivated **	BG/DE	Rhodopes mt.; German seeds	0.2	72	28	2.6	[36]
**81b**	Cultivated **	BG/UA	Rhodopes mt.; Ukrainian seeds	0.3	75	25	3.0	[36]
**81c**	Cultivated **	BG/AT	Rhodopes mt.; Austrian seeds	0.4	76	24	3.2	[36]
**82a**	Cultivated **	BG/DE	Vitosha mt.; German seeds	0.4	82	18	4.4	[36]
**82b**	Cultivated **	BG/UA	Vitosha mt.; Ukrainian seeds	0.5	77	23	3.3	[36]

* Accessions **34**–**35** were from Spain and accession 36 from the former Yugoslavia but without specification of the exact location [26]. ** Accessions **81** and **82** were plants cultivated from seeds originating from Ukraine, Germany and Austria [36]. ^a^ This accession Sufoio, although having an RHD slightly below 1, was classified as having a HEL chemotype by the authors [35]. Red and blue color denotes the HEL- and DH-chemotypes, respectively, as in Figure 3.

In addition to these various reports indicating that the HEL chemotype is a rather stable feature in the CEA, it has also been demonstrated, in at least two cases, that there may be exceptions, i.e., that flowers of *A. montana* growing in Eastern Europe may indeed also accumulate more DH than HEL derivatives [34] and, on the other hand, that the flowerheads of certain populations of Spanish *A. montana* display an STL profile with a higher content of HEL than DH compounds [35]. The latter study by Perry et al. [33] investigated 16 accessions of Spanish *A. montana* from three different types of habitats (meadows (n = 5), peat bogs (n = 8), heathland (n = 3)) at varying altitudes in Spain. The authors used an HPLC-UV method detecting at 225 nm but with their own internal standard and response factors. It was found, quite surprisingly, that accessions from three investigated heathland areas at high altitude (1330–1460 m) showed a chemotype dominated by HEL derivatives. The mean RHD in the three accessions was 1.6 (0.93–2.63), while the populations growing in the other two types of habitats and at lower altitude (420–1215 m) showed a DH chemotype, as that reported by Willuhn et al. [25], with a mean RHD of 0.12. Interestingly, the chemical differences between the mentioned accessions were also reflected in the pattern of phenolic constituents where the heathland samples also more closely resembled the Central European chemotype. However, the authors could not unequivocally assign the accessions based on morphological characteristics to the subspecies *montana* and *atlantica* sensu de Bolos y Vayreda [18] (see above), but the population with the highest HEL content was morphologically more similar to the former and the biometric features of some accessions with higher DH content more closely resembled the latter subspecies. The results of the Perry study [35] were later on combined, along with further chemical analyses, by Vera et al. [21,22] with their data from genetic analyses (compare Section 2.2 above), which indicated that there is a correlation of the population genetic groups with the two different chemotypes and it, hence, appears certain that the HEL chemotype does exist in certain Iberian populations. It is interesting to note that the authors presented evidence that the genotype corresponding to the DH chemotype may represent the ancestral form of *A. montana* [21].

On the other hand, in a very recent study on the influence of various environmental factors on the STL pattern of *A. montana* growing in the Apuseni mountain area in Romania, Greinwald et al. [34] reported an STL pattern dominated by DH derivatives for an Eastern European population. In this investigation, using the HPLC quantification method with individual correction factors by Willuhn and Leven [26], some differences in STL concentrations were found in flowerheads harvested in various grassland locations in the mentioned region, differing mainly in soil composition (siliceous vs. calcareous underground) as well as some other environmental factors. However, the average RHD values were 0.35 in the case of the former and 0.41 in the case of the latter soil type so that there appeared to be no significant difference in this parameter related to the soil composition. Thus, a DH-dominated chemotype of *A. montana* is not entirely confined to Iberic populations but obviously also exists in certain locations in Eastern Europe. Thus, a DH chemotype, though probably rarely, may also be found in *A. montana* far east and north of the Iberic peninsula. Clearly, this latter report [34] currently represents a singular finding, and it cannot be assessed on these grounds how frequently a DH chemotype will be encountered outside Iberia.

#### 2.2.4. The Occurrence of Different STL Chemotypes Is Not Confined to Different Populations

The question regarding the reasons for the major chemical differences between most CEA and SPA populations has been addressed various times. It would appear straightforward to explain the existence of different STL chemotypes with a fixed genetic difference also manifested in the existence of two geographically and morphologically distinguishable phenotypes. This would conveniently support the equivalence of the observed chemotypes with the proposed phenotypical subspecies [18], now also supported through investigations of the genotype [20,21] (i.e., subsp. *atlantica* = SPA = DH chemotype with RHD < 1, subsp. *montana* = CEA = HEL chemotype with RHD ≥ 1). However, the truth, as is often the case, appears to be more complicated than reflected by this simple a scenario. This becomes clear from the two examples just mentioned, where populations with the HEL chemotype occur in Spain [35] and where a DH chemotype is reported from Romania [34]. It becomes even more obvious in existing reports on the STL pattern of *A. montana* during plant development and ontogeny [36,37,38,39]. It was shown by Douglas et al. [37] (using the same analytical method as in [35]), that, in spite of an increase in the total STL amount in flowerheads of CEA (grown from European seeds and cultivated in New Zealand) with flowering time/stage within one vegetation period, no significant change occurred in the chemotype dominated by HEL-type STLs. Very interestingly, however, Todorova et al. [36], using a GC method with santonin as an internal standard, reported on a rather striking phenomenon where the STL chemotype did change over time within the same population: flowerheads of *A. montana* plants obtained via in vitro propagation of nodal sections from in vitro seedlings and then cultivated in proving fields in Romania, irrespective of their germplasm origin in Germany or Ukraine, displayed a change from a DH chemotype (RHD ≈ 0.3–0.4) in the second year of cultivation to an HEL chemotype (RHD ≈ 1–3) in the third year. Flowerheads of plants obtained directly from seeds (same origin as in vitro propagated plants) and grown in the same environment did not show this phenomenon, i.e., had the expected HEL chemotype (RHD ≈ 2–5) in the second year (first year in which flower samples were taken) [36]. These observations might be related to earlier observations made by the author of the present review with the leaves of cultivated *A. montana* [38]. Young leaves, growing from plants (three different accessions of proven CEA chemotype) in springtime of their third year in the proving field, first sampled 17 days after first sprouting, displayed an STL pattern (determined by GC according to [26]) with a high excess of HEL derivatives (RHD ≈ 4–7), which then rapidly changed within about 20 days of further leaf development to a pattern consisting almost exclusively of DH derivatives [38]. The latter then remained constant for the rest of the vegetation period, including the time of flowering (last sample taken 14 weeks after first sprouting). It was concluded from these observations that the change in the STL pattern over time must be due to an increase in the activity (or expression level) of a biogenetic enzyme, i.e., a hydrogenase converting HEL to DH derivatives, with leaf age (i.e., during leaf growth/differentiation) [38].

#### 2.2.5. Biogenetic Origin of Different STL Chemotypes

The biosynthetic sequence of STLs was proposed by early authors [40,41] and, in part, experimentally proven later [42] (see Figure 4). The initial cyclization of farnesyl diphosphate to germacrene A is followed by oxygenation of one of the carbon atoms in the isopropenyl side chain to a carboxy group. After hydroxylation at C-6 or C-8, the α,β-unstaurated lactone with an exocyclic double bond between C-11 and C-13 can then be formed. 11,13-Dihydro-STLs must then be formed via hydrogenation from the Δ^11,13^ unsaturated precursors. This conversion has been proven experimentally in the case of the simple germacranolide costunolide, which is converted by an oxygen-independent enoate reductase in an NADPH-dependent manner to 11(*S*),13-dihydrocostunolide [42]. The biosynthetic sequence to helenanolide-type STLs can be conceived to proceed, as shown in Figure 4 [40,41,42], where the transformations from the germacranolide precursor via a guaianolide intermediate to the pseudoguaianolide skeleton of the helenanolide are omitted for simplicity.

The hydrogenation of the helenalin derivatives to their 11α,13-dihydrohelenalin congeners (or the corresponding conversion at an earlier stage of biosynthetic precursors) would require at least one reducing/hydrogenating enzyme of the same type as responsible for the transformation of costunolide to 11,13-dihydrocostunolide [42]. It is not known (and not important for the present consideration) at which stage(s) of biosynthesis this enzyme can exert its activity (i.e., immediately on the germacrenolide, on an intermediate, on helenalin itself, or at various stages). It is straightforward to assume that it is this same enzyme that is responsible for the temporal changes in STLs in leaves [38] and in the flowerheads of in vitro propagated plants [36], as well as for the existence of two different chemotypes on CEA and SPA. Even in the absence of direct experimental proof for the existence of this enzyme in *A. montana*, it can be plausibly hypothesized that the activity or level of expression of this hydrogenase is higher in the flowerheads of most populations of SPA than in those of CEA. The reason for this differential activity most likely lies in differential levels of gene expression: the conversion of HEL to DH is not exclusive (and usually not entirely complete) in the DH chemotype but takes place to a different extent in both chemotypes. The extent of conversion (i.e., enzyme expression/activity) varies in each of the chemotypes (e.g., RHD values varied from >13 to ≈1 in CEA and from 0.18 to 0.02 in SPA [25]), and it has been observed that it can even vary over time in the same population’s leaves [38] as well as flowers [36]. Highly interesting data on the spatial and temporal distribution of STL biosynthesis in the flowerheads of *A. montana* cv. Arbo (along with data on a wild accession from Poland (i.e., both CEA) and with *A. chamissonis*, a North American species) were very recently published by Parafiniuk et al. [39]. The authors, based on LC-MS analyses, recorded the accumulation of H- as well as DH-type STLs in various parts of the flowerheads (buds, upper and lower parts of ray and disk flowers, pappus, receptacle, peduncle) at various flowering stages (bud, beginning flowering, full flowering and end of flowering stage) [39]. While the RHD was <1 at the budding stage in disk flowers and the receptacle and only 3.5 in ray flowers, it quickly changed to values >> 1 during the flowering stages in all parts of the flowerhead of cv. Arbo (data extracted from Tables S3–S6 of [39]). It would be extremely interesting to compare these results with analogous data from an SPA accession.

Thus, detailed space-/organ-/tissue- and time-resolved analyses of the STL pattern in direct comparison with the transcriptome and taking into account various parameters, including geographic origin, other environmental factors as well as the investigated plants’ ontogeny and developmental stage would be highly desirable to fully solve the enigma of why different chemotypes exist in *A. montana*.

## 3. Pharmacological Evidence

### 3.1. Wanted Pharmacological Effect: Anti-Inflammatory Activity—Implications for Drug Efficacy

As briefly mentioned above, STLs have long been accepted as the major pharmacologically relevant constituents in Arnica, which are held (and have in many cases been proven) responsible for the majority of the drugs’ therapeutic effects, including unwanted ones [1,3,8,12,13,14,15,43]. Since the flowerhead drug and its preparations are used—at least in science-based phytotherapy—against conditions related to inflammatory processes, the anti-inflammatory activity of STLs will be the main focus of this section.

Even in early studies dating back to the 1970s, HEL and related STLs were found to be more active than DH and other congeners without the 11,13 double bond (for reviews see, e.g., [1,3,8,12,13,14,15]). Briefly, HEL was found to be far more effective than various congeners with the hydrogenation of the 11,13 double bond in in vitro assays related to inflammatory processes such as the inhibition of liver cathepsin and acid phosphatase [44]. Most prominently, HEL showed almost seven times stronger anti-inflammatory activity than DH in vivo in the carrageenan-induced rat paw edema assay [45]. The analgesic activity of HEL was also proven to be more than five times stronger than that of DH in the acetic-acid-induced writhing reflex assay, serving as a model for inflammatory pain [45]. Various attempts were made in earlier years to explain the structure–activity relationships and the mechanism of action of the bioactivity of HEL and congeners as well as of Arnica preparations. It was generally accepted that the anti-inflammatory activity, in the same way as most other bioactivities of these compounds, such as antimicrobial activity and cytotoxicity, is related to the presence of the α,β-unsaturated carbonyl centers and it was also demonstrated various times that HEL as a bifunctional Michael acceptor generally has stronger activity than either 11,13 or 2,3 dihydrohelenalin derivatives (e.g., [44,45]; for reviews, see, e.g., [15,46,47,48]).

In a series of seminal investigations conducted by the group of I. Merfort and coworkers, in cooperation with the present author, during the late 1990s to early 2000s, the inhibitory activity of helenalin and congeners on the nuclear transcription factor NFκB was discovered [49,50,51,52,53,54]. This transcription factor controls the expression of various pro-inflammatory genes/proteins with relevance in immunological and inflammatory processes, such as cell adhesion molecules, immunoreceptors, cytokines (e.g., TNF-α, IL-1β and others), as well as enzymes (e.g., Cox-II, iNOS). The finding that HEL and congeners, like some other STLs, inhibit this important inflammatory switch, thus, offered a new and comprehensive explanation for the anti-inflammatory effects of Arnica and its STLs (for a review, see, e.g., [10,12,13]). Helenalin, initially reported to suppress the degradation of the inhibitory protein τκBα and, thereby, to hinder activation of NF-κB [49], was later shown to directly inhibit the binding of NFκB to its DNA motif and its capability to induce the transcription of pro-inflammatory genes [50,51,52]. HEL was first predicted on grounds of molecular models [52] and later shown experimentally [53] to bind covalently to a particular cysteine residue (C38) in the DNA binding interface of the transcription factor’s p65 subunit. The ability of the factor to bind to DNA and, thus, to induce transcriptional activity is thereby blocked [53]. It was demonstrated that DH also inhibits NF-κB but does so with only about 20-fold lower potency than HEL [52].

A study of very high relevance with regard to the present review’s central question was conducted by Klaas et al. in 2002 [54]. Investigations comparing Arnica tincture prepared from flowers of CEA (*A. montana* cv. ARBO, HEL chemotype) with tincture made from SPA (DH chemotype) resulted in the finding that the inhibition of NF-κB was about twice as strong in the case of the former than the latter [54]. This was in agreement with the much stronger (about 2- and 5-fold, respectively) inhibitory effect of the CEA tincture on the expression of the cytokines IL-1β and TNF-α, both of high relevance in inflammation and expressed under the control of NF-κB [54]. In addition to this, the tincture prepared from CEA also displayed around a 2.5-fold stronger inhibitory effect than the SPA tincture on the DNA binding of another transcription factor with high importance in inflammation, namely, NFAT (nuclear factor of activated T cells) [54]. Inhibitory activity on NFAT could also be proven for pure HEL by other authors [55], so that this effect of the tincture can thus be attributed to the STL as well. Klaas et al. also demonstrated that esters of DH with α,β-unsaturated carboxylic acids (i.e., tiglate and methacrylate, DHTG, DHMA) are about twice as active as NF-κB inhibitors than unesterified DH or its acetate DHAC [54]. However, this level of activity is still about 10-fold lower than that of unesterified HEL and 5-times lower than in the case of the isobutyric acid ester, HELIB [52]. DHMA, as the main constituent of the SPA tincture, was also proven to be slightly more active in vivo against croton oil-induced mouse ear edema than the acetate DHAC [54]. Unfortunately, a direct comparison of in vivo efficacy with the corresponding HEL esters was not reported, and there was also no comparison of the two tinctures’ in vivo efficacy.

Similar results were obtained by the same group when Arnica tinctures prepared from CEA and SPA Arnica flowers were reported to modulate the activity of the matrix metalloproteases MMP1 and MMP13 in human and bovine chondrocytes by inhibiting the transcription factors AP-1 and NF-κB [56]. Significantly higher amounts of a tincture obtained from SPA than of two investigated tinctures made of CEA flowerheads were needed to reach the same effects, in spite of the higher total STL content of the former [56].

In addition to the inhibition of these transcription factors, further mechanisms of action have been shown for HEL-type STLs, which can also be considered important for the overall anti-inflammatory activity. Thus, it has been shown that HEL is an inhibitor of Leukotriene biosynthesis [57]. The STL inhibited 5-lipoxygenase as well as leukotriene C4 synthase in human thrombocytes at concentrations similar to those required for NF-κB inhibition, and it was concluded that the effects on leukotriene synthesis might, hence, well be involved in the overall anti-inflammatory action of the STL. In the same study, DHAC was also tested and turned out to be a much weaker inhibitor of the mentioned pro-inflammatory enzymes [57].

Much more recently, and possibly more importantly, it was discovered that STLs like HEL and some other types inhibit the activity of yet another transcriptional system, namely, the MYB-C/EBPβ-p300 transcription module (for a recent review, see [58]). STLs were initially reported to inhibit the activity of the transcription factor cMyb [59,60], but it was later found in studies with HELAC that the observed effects were actually due to the inhibition of another transcription factor cooperating with cMyb, namely C/EBPβ (CCAAT box/enhancer binding protein beta). Even more precisely, HELAC inhibits the interaction of C/EBPβ and the coactivator p300 [61,62], which then also abrogates the expression of cMyb-dependent genes. Apart from the MYB-C/EBPβ-p300 module’s eminent importance in cell differentiation and tumorigenesis, it is known that the transcription factor C/EBPβ also plays important roles in inflammation ([58,62] and literature cited there). In fact, another name for C/EBPβ is “nuclear factor of interleukin-6” (NF-IL6), due to its role in controlling the expression of this important cytokine, with high relevance in inflammation [63]. For HELAC, it was demonstrated that the IC_50_ concentration for C/EBPβ inhibition is at least 10-times lower than that required for NFκB inhibition so that the new mechanism may be even more important for the anti-inflammatory action of Arnica STLs than their effect on NF-κB [62]. Among a broad variety of STLs of various types tested for structure–activity relationships, HEL and several of its esters all showed much higher activity than DH esters: the IC_50_ values for cMyb inhibition (which later turned out to be C/EBPβ inhibition) determined for HELAC and DHAC were 0.7 and 18 µmol/L, respectively [60]. Thus, also in this new mechanistic aspect, shedding further light on the Arnica STLs’ anti-inflammatory action, HEL congeners turned out to be far more effective agents than DH analogues.

A relatively limited number of clinical studies on the therapeutic efficacy of Arnica preparations exist. The existing studies were summarized in the ESCOP monograph [15] and the EMA assessment report [16]. To the author’s knowledge, no clinical study so far has compared the efficacy and/or safety of Arnica preparations obtained from flower drugs representing the CEA and SPA chemotypes.

However, based on many results from biological assays related to the anti-inflammatory potency of HEL and DH and their derivatives, including such (few) preparations of HEL and DH chemotypes being directly compared, it can be concluded that Arnica preparations with a higher content of HEL derivatives (i.e., RHD > 1, corresponding to the typical CEA)—at least in the case of an approximately equal total STL content—will be superior in efficacy to those that are characterized by a dominance of DH derivatives (typical SPA type).

### 3.2. Unwanted Effects—Sensitizing and Acute Irritant Potential—Implications on Drug Safety

#### 3.2.1. Contact Allergenic Potential

Many plants of the family Asteraceae (=Compositae), including Arnica, are known to cause contact allergy [43,64,65,66]. STLs are mainly held responsible for this condition known as Compositae allergy, since many of them contain reactive electrophilic structure elements that can form covalent bonds with nucleophilic groups in biomolecules. Reaction with skin proteins after external contact is thought to convert the STLs, acting as haptens, into full antigens, which are then internalized by antigen-presenting cells (APCs) of the skin, i.e., Langerhans cells. The latter, after processing, presents fragments of the modified proteins to T cells, which are then activated to initiate an inflammatory immune response. Upon recontact, the response is mediated by memory T cells and characterized by the delayed onset of inflammatory symptoms, as typical for type IV contact hypersensitivity [43,64,65,66]. Contact allergies of this type have been described for many Asteraceae, and it is known that there is a considerable degree of cross-sensitivity between different plants of the family and/or their STLs (for a review with further literature, see [48,67]), but there are only sparse quantitative data on the frequency of Arnica allergy. While Hausen and Vieluf, in their standard book, classify the sensitization potential of Arnica as “high” [66], Willuhn [43] cited data from various investigations, in which the frequency of positive skin reactions to Arnica tincture in the epicutaneous testing of eczema patients was reported. Of these patients, 4.3% (150 of 3480 patients) showed a positive reaction to Arnica tincture. This does not imply that the positive persons had been sensitized directly to Arnica since the potential for cross-sensitivity is high in the case of Asteraceae and their STLs [43,64,65,66,67] (further work on Arnica allergenicity in general is summarized in [15]). According to more recent data on the exposition of 1280 probands, the unwanted skin reaction (i.e., allergic contact dermatitis) can occur in 1:100 users of Arnica [16].

It is commonly accepted that allergenic STLs, including those of Arnica, undergo Michael-type addition reactions with proteins (especially with Cysteine SH groups) with their electrophilic α,β-unsaturated carbonyl structures (“enones”) and, thereby, not only cause most of their wanted pharmacological effects, such as the inhibition of NF-κB (see above, Section 3.1), but also elicit the immunological reactions described above. A particularly frequent structure element of this kind is the α-methylene-γ-butyrolactone (ML) group (for a review, see [48]), and it has been demonstrated in many cases that this feature can be essential for the allergenicity of an STL. Among the STLs of *Arnica montana*, only compounds of the HEL series possess this structure element. However, other enone moieties, including the cyclopentenone (CP) group occurring in HEL as well as DH derivatives, are also able to react with SH groups [68]. The DH derivatives must, hence, also be considered as potential haptens able to elicit contact hypersensitivity, and this was also proven experimentally [43,69]. However, HEL derivatives as bifunctional Michael acceptors with two different reactive centers, an ML and CP moiety, would possibly represent more potent haptens than the DH congeners featuring only the latter so that a DH-dominated chemotype might be expected to be less allergenic. This was indeed assumed in various reports, but there appears to be little experimental evidence to support the assumption. One such result pointing in this direction was reported by Lass et al. [70], who found that Arnica tincture prepared from CEA with an HEL chemotype was able to cause contact hypersensitivity in acutely CD4-depleted mice, while this was not the case with tincture from SPA with a DH chemotypes. However, this result was only obtained in an artificial mouse model. Interestingly enough, it was not possible to induce contact allergy in normal mice with *either* tincture or even with isolated HEL and DH esters (HELIB, DHIB, DHMA), and the authors presented good evidence that the anti-inflammatory activity of the STLs indeed counteracts their sensitizing potential. Therefore, the authors classified Arnica as a weak allergen [70]. In a later report, these authors also presented evidence that the tincture with a DH chemotype, but not the HEL chemotype, induces a somewhat elevated expression of the anti-inflammatory cytokine, interleukin-10 (IL-10) [71], which would point in the same direction. However, in view of the many reports on significantly stronger anti-inflammatory effects by HEL derivatives (see Section 3.1 above), it is unclear whether a more favorable benefit/risk ratio could really be expected from an Arnica preparation of the DH chemotype, even more so since, in fact, neither tincture caused allergy in wild-type mice [70]. It was even shown in a study with human patients with a known Arnica allergy that a higher ratio of these individuals (two of eight) reacted to DHMA than to a helenalin ester (HELIB, zero of eight, with one patient showing an irritant rather than allergic reaction, though) [72]. However, this was a very small sample and might, thus, not be representative. Nevertheless, the low response rate to either of the two STLs, on the one hand, raises the question of which other compound(s) in addition to STLs may be co-responsible for the allergenicity of Arnica. It is possible that thymol derivatives, common constituents of the essential oil of *Arnica* species, including *A. montana* [24], may be involved in this additional allergenicity. This is supported by a study in which 10-acetoxy-8,9-epoxy-thymol-isobutyrate was identified as the contact allergen of *A. sachalinensis*, an east Asian *Arnica* species not containing any STLs at all [73]. On the other hand, the mentioned results [72,73] indicate that a chemotype dominated by DH derivatives such as typically encountered in SPA would not per se promise to cause less allergic reactions in Arnica-sensitized persons than the HEL chemotype of CEA.

#### 3.2.2. Acute Skin Irritant Potential

In addition to the immunological potential to cause and elicit contact allergy, it has also been reported that Arnica may cause acute skin irritation, especially after topical exposure to higher concentrations [43,47]. This is the main reason why it is recommended to use Arnica tincture only after dilution (1:3–1:10) with water [14,15,16,17]. It seems to be unclear up to the present whether this unwanted irritant reaction—as assumed by Willuhn [43]—is caused primarily by components of the essential oil or whether it is also due to the STLs. The latter would be supported by data from an American group, who demonstrated that HEL, as some other STLs, potently causes mast cell degranulation and histamin release while tenulin, a HEL analogue, possessing only the CP but not the ML moiety and, thus, structurally related to DH, almost inactive in this respect [74]. Even though none of the DH derivatives occurring in Arnica were investigated in this study, this finding might point towards a lower acute irritant potential of Arnica with a chemotype containing less HEL and more DH derivatives. However, this would have to be proven experimentally through direct comparison of the activity of at least one pair of homologous HEL and DH esters on mast cells.

Interestingly, results were reported for two DH esters that would support the idea of a lower irritant and/or allergenic potential of the DH chemotype. After isolation from another Asteraceous plant, *Centipeda minima*, DHIB and DHSE even showed an inhibitory effect on histamin release from mast cells [75,76].

At this point, the study by Lass et al. mentioned again. These authors demonstrated a somewhat more favorable effect of a DH-type tincture on anti-inflammatory IL-10 expression [71], which would also support a more favorable behavior with respect to acute irritant activity.

All this said, it is interesting to note that the skin toxic and irritant potential of Arnica tincture was very recently reported to be insignificant [77]. In the course of a study on the efficacy of Arnica tincture against cutaneous Leishmaniasis, the acute and chronic skin toxicity was investigated in vivo (mouse), and the corrosive and irritant potential was tested in a reconstituted human epidermis (RhE) model. Arnica tincture of the HEL chemotype (commercial origin, total STL content 0.05% (*m*/*v*), determined via a fully validated LC-MS method [78]; RHD = 1.9, i.e., representing a CEA type) did not show any positive reactions in any of these tests. It is worth mentioning that the tincture was used in undiluted form in these experiments. The acute irritant potential of Arnica tincture produced from flowerheads of CEA, on the background of these investigations, can be considered relatively low in spite of the considerable content of HEL esters. Although it is not possible to exclude the possibility, it appears uncertain on this background whether a DH chemotype could offer an appreciably lower potential for acute skin irritation.

## 4. Conclusions and Future Directions

The present situation of Arnica as a drug and its preparations, in spite of its long tradition as a successful herbal remedy, is far from satisfactory. It is firmly established on the grounds of a large body of evidence from many chemical investigations that “Arnica flower”, in terms of chemical profile and, thus, pharmacological potency, may mean different things. With respect to pharmaceutical/medicinal use, it is unimportant whether a taxonomic subdivision in two subspecies, *montana* and *atlantica*, is correct and accepted in the sense of systematic botany. The existence of two chemotypes within the species is an undeniable fact. This fact is obviously being ignored by the Ph. Eur. (i.e., by the European legislative authorities). Two rather different herbal drugs with conspicuous differences in chemical constituents and, thus, pharmacological potency as well as, possibly, a different level of risk are called the same name and treated as the same thing, which cannot be considered satisfactory in terms of the pharmaceutical quality, efficacy and safety of Arnica preparations.

On the grounds of pharmacological evidence, it would appear reasonable to use *A. montana* of the HEL chemotype for a genuine medical purpose where it clearly is the anti-inflammatory potency that counts. In this regard, it should be mentioned again that an easily cultivable clone, *A. montana* Arbo, is available, which displays an HEL chemotype. The DH chemotype, on the other hand, could possibly be utilized more confidently in a cosmeceutical context (whether it has a lower risk of allergy or not) where the pharmacological potency is less important.

In view of the presented evidence, it is now necessary to accept the facts and implement a distinction between a helenalin- and a dihydrohelenalin-rich chemotype of *Arnica montana* in the pharmacopoeia monograph. In this implementation, care should be taken of the mentioned analytical shortcoming of the Ph. Eur. method by prescribing at least a distinction between the two groups of STLs, which does not represent a technical problem. The distinction between two chemotypes must be demanded, given the background of all evidence and current knowledge presented in this review, for the sake of pharmaceutical quality, efficacy and safety, despite the fact that not all relevant questions are answered. It will be of significant interest to study, in more detail, the origin/reasons for the occurrence of the large quantitative differences between HEL and DH derivatives in the two chemotypes or putative subspecies. Such studies would best include very thorough time- and organ-resolved analyses of the STL pattern in *A. montana*, best grown under controlled environmental conditions, paralleled by analyses of gene expression/transcriptome in the same plants. Furthermore, detailed pharmacological comparisons at the in vitro as well as in vivo and, optimally even at the clinical, levels between the two chemotypes will have to be conducted in future in order to arrive at valid quantitative measures for the efficacy and safety of the two Arnica flower drugs of different provenance.

## 5. Materials and Methods

Please note that this review is not meant as a comprehensive review on the topic “Arnica” but is focused on the botanical, chemical and pharmacological differences between the two *A. montana* (chemo)types under debate. The current work is, hence, largely based on the author’s literature collection on Arnica accumulated over the years. Additionally, literature searches in PubMed [79] (https://pubmed.ncbi.nlm.nih.gov (accessed on 27 July 2023)) and SciFinder [80] (https://scifinder-n.cas.org/ (accessed on 27 July 2023)) were conducted in July 2023. In order to exclude articles dealing with Arnica as a homeopathic remedy, the key word “Arnica montana NOT homeop*” was used.

In the case of SciFinder, the 1186 initial hits were filtered by SciFinder sections “Pharmaceuticals”, “Pharmacology”, “Pharmaceutical analysis”, “Plant Biochemistry”, “Toxicology”, “Biological Chemistry: Pharmacology” and “Biological Chemistry: Botany”, leaving 265 search results. The PubMed search yielded 402 hits, which were not further filtered.

The remaining titles and abstracts (402 + 265) were then evaluated for direct relevance to the topic of the review, i.e., comparisons/differences between *Arnica montana* chemotypes or Arnica of different geographic origins.

The author apologizes in advance for possibly having missed any relevant pieces of literature information and hereby also expresses his advance gratitude for being notified if any should be detected.

## Figures and Tables

**Figure 1 plants-12-03532-f001:**
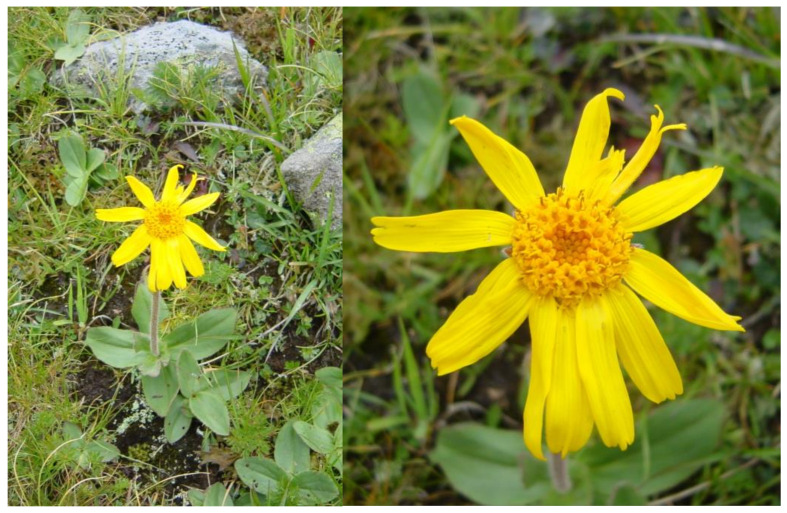
*Arnica montana* L. (**Left**): flowering whole plant in its natural habitat; (**right**) closeup of the flowerhead.

**Figure 2 plants-12-03532-f002:**
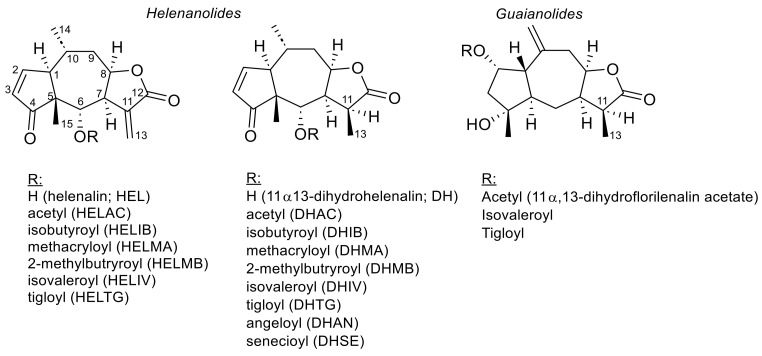
Structures of sesquiterpene lactones known from *Arnica montana*.

**Figure 4 plants-12-03532-f004:**
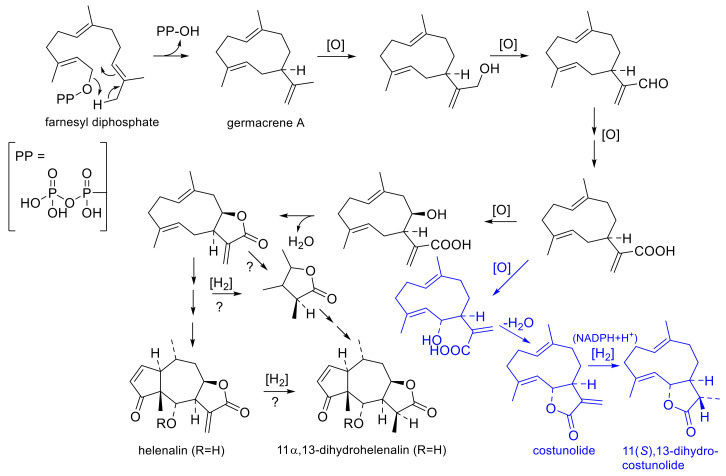
Biosynthesis of sesquiterpene lactones following [40,41,42]. The existence of an enzyme hydrogenating the 11,13-double bond of costunolide in a stereospecific manner to 11(*S*),13-dihydrocostunolide (shown in blue color) was experimentally proven in Cichory root [42]. That a corresponding enzyme is involved in the biosynthesis of 11α,13-dihydrohelenalin and -derivatives, appears likely on grounds of observations of temporal changes in STL pattern in *A. montana* leaves [38] as well as flowerheads [36]. It is unclear, at present, at which stage(s) of biosynthesis this hydrogenation takes place; the corresponding reactions are therefore marked “?”.

**Table 1 plants-12-03532-t001:** Response factors K_f_ for the HPLC quantification of various Arnica STLs as determined by Leven and Willuhn [26] and their reciprocal values in comparison with the substance specific correction factor SCF of Ph. Eur. [4,11] for DHTG.

STL	K_f_ [26]	1/K_f_	SCF [4,11]
DH	0.80	1.250	
DHAC	0.85	1.176	
DHTG ^1^	0.88	1.136	1.187
DHIV	0.87	1.149	
HEL	2.20	0.455	
HELAC	1.23	0.813	
HELTG ^1^	1.15	0.870	

^1^ Example: If the content of an Arnica tincture were determined as 0.050% STLs calculated as DHTG according to Ph. Eur. (SCF = 1.187), the same tincture would only be calculated to contain 0.037% STL if the SCF for HELTG = 0.870 were used. If the peak area of STLs in this sample were made up of, e.g., 50% HEL and 50% DH derivatives, the “true” content, calculated with the proper SCF values of HELTG and DHTG, respectively, would be 0.018% HEL and 0.025% DH = 0.043% total STL. In case of 90% HEL: 10% DH peak area ratio, use of the appropriate SCFs would lead to 0.033% HEL and 0.005% DH derivatives, i.e., a total content = 0.038%. The value of 0.05% determined according to Ph. Eur. would thus be 16% and 31% too high, respectively, in the two examples.

## Data Availability

Not applicable.

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
