# Peer review of "Arnica montana L.: Doesn’t Origin Matter?"

_plants, 2023, doi:10.3390/plants12203532_

Round 1

Reviewer 1 Report

The manuscript addresses the question of the relationship between the geographical origin of Arnica montana flowers and relevance to the medicinal use of herbal medicine and the pharmaceutical quality, efficacy and safety of its products.

Some points to consider in the review:

1) In the introduction, the author could add a figure with geographic distribution, since the issue of geographic origin is addressed.

2) He could write a paragraph on taxonomy. Strange to see that the botanical family is not mentioned in the introduction.

3) Regarding chemical composition, the author could make a table listing chemical compound, percentage, population of origin. In this way, it would be possible to infer chemical variation and place of origin.

4) Considering that there is a lot of information about Arnica montana and that it is consumed worldwide, is there any approach to genetic diversity, cultivation and reproduction approaches?

5) Furthermore, to improve the review, it could address cultivation and climatic conditions.

Author Response

Reviewer 1

The manuscript addresses the question of the relationship between the geographical origin of Arnica montana flowers and relevance to the medicinal use of herbal medicine and the pharmaceutical quality, efficacy and safety of its products.

Some points to consider in the review:

1) In the introduction, the author could add a figure with geographic distribution, since the issue of geographic origin is addressed.

Answer: I thank the reviewer for this good suggestion! A map with the geographic distribution and the approximate locations of over 80 analyzed accessions has been generated and is presented as Figure 3. In this map, I also included a color code indicating the chemotype. This map will then work best in combination with a Table generated as suggested by this reviewer under 3) below.

2) He could write a paragraph on taxonomy. Strange to see that the botanical family is not mentioned in the introduction.

Answer: The reviewer may have overlooked that the systematic placement of A. montana within the genus Arnica is described in section 2.1. In this paragraph, there is also a rather comprehensive description of the infraspecific taxonomy (postulated subspecies etc.), which is very important in the context of this review. The systematic placement of the genus Arnica within the subtribes and tribes of subfamily Asteroideae, which has been a matter of much debate over decades, is not mentioned in the current review because this is not relevant for the current question to be answered by this review.

I do agree with the reviewer that the family, of course, needs to be mentioned also in the introduction. It was added.

3) Regarding chemical composition, the author could make a table listing chemical compound, percentage, population of origin. In this way, it would be possible to infer chemical variation and place of origin.

Answer: I thank the reviewer for this very good suggestion. The map was compiled and is now presented as Table 2. The amounts of HEL and DH derivatives as well as their ratio (RHD values) have been included. The map then works in combination with Figure 3 where the geographic locations of origin are shown (see 1) above).

4) Considering that there is a lot of information about Arnica montana and that it is consumed worldwide, is there any approach to genetic diversity, cultivation and reproduction approaches?

Answer: It is a protected plant and at the same time difficult to cultivate. However, attempts to find a more easily cultivatable clone (cv. ARBO) were successful already a long time ago. I have added a sentence on this in section 1.

5) Furthermore, to improve the review, it could address cultivation and climatic conditions.

I would rather not distract the review’s focus too far from the actual topic and question to be answered. I hope the reviewer will accept that I have not added further details on these issues which may certainly be an interesting topic for an article more specialized in the cultivation of A. montana.

I thank this reviewer for the time and thoughtful effort spent to help me improve this manuscript!

Reviewer 2 Report

I would recommend changing the title of the review. The origin of Arnica is discussed briefly. 

I would recommend a brief description of the medicinal use of Arnica throughout history. Are there new medicinal uses for this plant species compared to centuries ago?

There is a review by Kriplani et al., 2017. Arnica montana L. - a plant of healing: review. The Journal of pharmacy and pharmacology, 69(8), 925–945. https://doi.org/10.1111/jphp.12724, with similar content to the current manuscript. The difference between this review in comparison to other reports should be explained in the introduction section. 

Section 2. Is the population of this plant species with any conservative concern? 

Section 3. Even if the author mentions that there are few reports, I recommend focusing this section on clinical trials with information on the plant extract or active compounds. 

I recommend adding a section about herbal products containing Arnica. Part of this information is mentioned in the introduction section briefly. Are these products safe for consumption? The information could be useful to the general population who might read this review article. 

No comments

Author Response

Reviewer 2

I would recommend changing the title of the review. The origin of Arnica is discussed briefly. 

Answer: I do not understand the reviewer’s suggestion. If a change of title is suggested, then how should it read? Which aspect should be changed and how? At the same time, I am sure the title is appropriate to reflect the content since the chemical and pharmacological differences in relation to the plant material’s geographic origin are not only discussed briefly but represent the core topic of this article. Therefore, I would rather not change the title.

I would recommend a brief description of the medicinal use of Arnica throughout history. Are there new medicinal uses for this plant species compared to centuries ago?

Answer: I do not wish to repeat too much of what has already been summarized in previous reviews and monographs and this is also not the topic here.

There is indeed a potential new area of us, i.e. against cutaneous Leishmaniasis which is currently in clinical study, which is briefly mentioned in section 3.2.2. I do not find it useful in the context of the present article to write more about this promising new indication. It is not directly related to the topic of this review and will be reserved to other publications.

There is a review by Kriplani et al., 2017. Arnica montana L. - a plant of healing: review. The Journal of pharmacy and pharmacology, 69(8), 925–945. https://doi.org/10.1111/jphp.12724, with similar content to the current manuscript. The difference between this review in comparison to other reports should be explained in the introduction section. 

Answer: To be honest, the review article suggested by the reviewer contains many errors and does not fully cover the relevant literature. In some instances, it does even contain false statements. Just one example: Chamissonlide and Mexicanin I are mentioned as constituents of Arnica montana which is wrong: the former occurs in A. chamissonis and some other Arnica species but NOT in A. montana; the latter is a constituent of A. acaulis. The original seminal work of the group of Willuhn et al., in which these and many other STLs were isolated and identified, is insufficiently cited in this review.

Most importantly, this suggested review -even though shortly mentioning some differences in chemical constituents in a rather rudimentary (and not entirely correct) manner- does neither address the existence of two different chemotypes related to geographic origin nor the possible existence of two subspecies at all. This is why I will not cite it because IF I would cite it, I would have to mention its shortcomings (a few more than summarized here) which is not necessary in the context of this present article.

Section 2. Is the population of this plant species with any conservative concern? 

Answer: There are many populations of this plant species and, as mentioned, it is a protected plant in Europe. Luckily, a more easily cultivable clone, registered as cultivar Arbo, was developed which is also available on the market. I have added a statement on this in section 1.

Section 3. Even if the author mentions that there are few reports, I recommend focusing this section on clinical trials with information on the plant extract or active compounds. 

Answer: Even if the reviewer asks for it, there are no clinical trials with information on the origin of the plant material and its exact chemical composition. In particular not a single clinical trial has compared the two chemotypes. Therefore, I am sorry that I cannot follow the reviewer’s request. This is simply due to lack of literature data.

I recommend adding a section about herbal products containing Arnica. Part of this information is mentioned in the introduction section briefly. Are these products safe for consumption? The information could be useful to the general population who might read this review article.

Answer: In fact, the relevant information about herbal products is already contained in section 1, where also the relevant monographs containing more details are cited. The safety “for consumption”, as mentioned there, is limited by the legal authorities to external use (all else is homeopathy and, as such, not to be commented in a scientific article). Further considerations about possible safety issues (allergy etc.) are summarized and discussed quite thoroughly in section 3.2. The author does not know what else should be added in a separate / new section.

I thank this reviewer for the time and effort spent.

Round 2

Reviewer 2 Report

The manuscript can be accepted for publication

No comments